# Improvement of Monacolin K and Pigment Production in *Monascus* by 5-Azacytidine

**DOI:** 10.3390/jof10120819

**Published:** 2024-11-26

**Authors:** Chan Zhang, Haijiao Wang, Qing Sun, Arzugul Ablimit, Huijun Dong, Congcong Wang, Duchen Zhai, Bobo Zhang, Wenlin Hu, Chengjian Liu, Chengtao Wang

**Affiliations:** 1School of Food and Health, Beijing Technology & Business University (BTBU), Beijing 100048, China; wanghj2023@163.com (H.W.); 13131535358@163.com (Q.S.); guzal_arzu@163.com (A.A.); 15233032630@163.com (H.D.); wangcongcong_111@163.com (C.W.); duchenz0117@163.com (D.Z.); 2Beijing Advanced Innovation Center for Food Nutrition and Human Health, Beijing Technology & Business University (BTBU), Beijing 100048, China; 3Beijing Engineering and Technology Research Center of Food Additives, Beijing Technology & Business University (BTBU), Beijing 100048, China; 4School of Science, Shantou University, Shantou 515063, China; bbzhang@stu.edu.cn; 5Guangdong Tianyi Biotechnology Co., Ltd., Zhanjiang 524000, China; hwl0769@163.com; 6Shandong Fanhui Pharmaceutical Co., Ltd., Jinan 271100, China; shandongfanhui888@163.com

**Keywords:** *Monascus*, 5-azacytidine, pigments, Monacolin K, HPLC

## Abstract

*Monascus* species are known to produce various secondary metabolites with polyketide structures, including Monacolins, pigments, and citrinin. This study investigates the effects of 5-azacytidine on *Monascus* M1 and RP2. The dry weight, red, yellow, and orange pigment values, and Monacolin K yield of both *Monascus* strains were measured, and their hyphae observed through electron microscopy. The experimental group showed higher dry weights and pigment values than the control group for both strains. However, Monacolin K production increased substantially only for *Monascus* M1. Electron micrographs revealed surface wrinkles and large protrusions in both strains after 5-azacytidine treatment. As a potent DNA methylation-promoting agent, 5-azacytidine is very useful for epigenetic and cancer biology studies and for studying secondary metabolism in fungi.

## 1. Introduction

*Monascus* species have been used as fermentation strains in food and pharmaceutical industries for more than 1000 years [1,2]. They are valued for their nutritional properties, research utility, and documented medicinal efficacy [3]. Secondary metabolites produced by *Monascus* include pigments [4], Monacolins [5], citrinin [6], and γ-aminobutyric acid [7]. Filamentous fungi, such as *Monascus*, have a high capacity to synthesize rich secondary metabolites, including hypolipidemic drugs such as Monacolin K [8]. Monacolin K exists in both acid and lactone forms. The lactone form needs to be converted to the acid form by the hydrolysis of carboxyesterase in humans. However, the structure of *Monascus* natural fermentation is mostly acid form, so it can directly play a lipid-lowering role without conversion. Monacolin K, also known as Lovastatin, inhibits cholesterol biosynthesis by competitively inhibiting 3-hydroxy-3-methylglutaryl-coenzyme A (HMG-CoA) reductase. The Monacolin K biosynthetic gene clusters in *Monascus* are well studied and share similarities with the synthetic genes involved in Lovastatin synthesis. With the growing demand for healthy, safe food products, *Monascus* sp. have garnered attention as a sustainable source of natural colorants. It is essential to develop simple and efficient methods to improve its production.

Both prokaryotic *Streptomyces* and fungi can heterologously express entire biosynthetic gene clusters [9] through the modification of culture conditions [10] or the overexpression of specific transcriptional regulatory genes that control secondary metabolite synthesis [11]. Altering endogenous and exogenous conditions during fermentation may induce epigenetic modifications, markedly enhancing the biosynthesis of microbial secondary metabolites.

DNA methylation is an epigenetic modification, and together with the expression of silencing genes in fungi, can be activated by DNA methyltransferase inhibitors [12]. These include 5-aza-2′-deoxycytidine and 5-azacytidine, which activate the hygromycin resistance gene in fungi [13] and the phleomycin resistance gene in *Phanerochaete chrysosporium* [14]. In *Aspergillus terreus*, *LaeA* regulates Lovastatin biosynthesis, while in *Aspergillus nidulans*, *LaeA* regulates both sterigmatocystin and penicillin biosynthesis, as well as the Lovastatin biosynthetic gene cluster [15]. The global regulator gene *LaeA* was isolated from *Monascus purpureus* M1 to create an overexpression construct. The resulting LaeA-overexpressing strain (L3) produced 48.6% more Monacolin K than the M1 strain [16]. DNA methylation and histone deacetylation are key mechanisms in the epigenetic modification of eukaryotic gene regulation [17]. Moreover, studies have shown that 5-azacytidine is closely related to the regulation of the LaeA factor [18]. As a type of DNA methylator, 5-azacytidine mainly affects the G1 phase of the cell cycle [19]. It is hypothesized that 5-azacytidine may influence the growth and development of *Monascus* during the G1 phase.

This study examines the effects of 5-azacytidine on two *Monascus* strains, M1 and RP2. A defined amount of 5-azacytidine was added to both strains during the G1 phase to evaluate its impact on the preservation of Monacolin K and to verify whether 5-azacytidine increased the secondary metabolite production of the two strains. In addition, we measured the dry weight of *Monascus* M1 and RP2; red, yellow, and orange pigment values; and yield of Monacolin K on days 2, 5, 8, 10, 12, 14, 16, and 18. We observed their hyphae using electron micrographs on day 8 to verify whether 5-azacytidine methylation affected the secondary metabolites of *Monascus*.

## 2. Materials and Methods

### 2.1. Strains and Materials

*Monascus* M1, a stable producer of Monacolin K, was obtained from the Chinese General Microbiological Culture Collection Center (Strain No. CGMCC 3.0568), Beijing, China. *Monascus* RP2 is a high producer of pigments (Strain No. CGMCC 18801). Both *Monascus* strains were maintained on potato dextrose agar (PDA) for 5 days at 30 °C. In order to verify the promoting effect of 5-azacytidine on Monacolin K and pigment, M1 and RP2 strains were selected for the experiment.

*Monascus* M1 was cultured in seed fermentation medium containing glucose 30 g/L, soy flour 15 g/L, MgSO_4_ 1 g/L, KH_2_PO_4_ 2 g/L, glycerol 70 g/L, peptone 10 g/L, and NaNO_3_ 2 g/L, at 30 °C and 200 rpm shaking for 48 h. Afterwards, the culture was transferred to a fermentation medium containing glycerol 90 g/L, indica rice flour 20 g/L, peptone 10 g/L, NaNO_3_ 5 g/L, MgSO_4_ 1 g/L, ZnSO_4_ 2 g/L, and KH_2_PO_4_ 2.5 g/L, and cultured at 30 °C and 150 rpm shaking for 48 h, and then at 25 °C and 150 rpm shaking for 18 days.

*Monascus* RP2 was cultured in seed fermentation medium containing indica rice flour 40 g/L, peptone 8 g/L, soybean meal 5 g/L, KH_2_PO_4_ 2 g/L, NaNO_3_ 2 g/L, and MgSO_4_·7H_2_O 1 g/L at 33 °C and 200 rpm shaking for 48 h. Afterwards, 5 mL of liquid was transferred to 50 mL fermentation medium containing indica rice flour 77 g/L, glucose 75 g/L, soybean meal 2 g/L, KH_2_PO4 0.5 g/L, NaNO_3_ 1.8 g/L, MgSO_4_·7H_2_O 1 g/L, and corn steep liquor 3.5 g/L, and cultured at 33 °C and 200 rpm shaking for 18 days.

PDA, glycerol, glucose, soybean meal, peptone, KH_2_PO_4_, NaNO_3_, MgSO_4_·7H_2_O, ZnSO_4_·7H_2_O, corn steep liquor, PBS, glutaraldehyde, isoamyl acetate, and ethanol were purchased from Beijing Aoboxing Biotechnology Company Ltd. (Beijing, China). 5-azacytidine was purchased from Biotopped Technology Co., Ltd. (Beijing, China).

### 2.2. Addition of 5-Azacytidine

Before fermentation, varying concentrations of 5-azacytidine were added to 50 mL of fermentation medium, with a concentration gradient of 1 g/L, 10^−1^ g/L, 10^−2^ g/L, 10^−3^ g/L, 10^−4^ g/L, 10^−5^ g/L, and 10^−6^ g/L. The yield of Monacolin K in the M1 strain and the red pigment in the RP2 strain on the fifteenth day were used to determine the optimal 5-azacytidine concentration.

After 2 days of fermentation, *Monascus* M1 and RP2 entered the logarithmic phase, suggesting that the G1 phase occurred within 48 h of culture in the seed fermentation medium. Thus, the optimal concentration of 5-azacytidine was added to 50 mL seed fermentation medium at 4 h intervals within 48 h. The yield of Monacolin K in the M1 strain and the red pigment in the RP2 strain on day 15 were used to determine the optimal addition time.

### 2.3. Determination of Pigment Value

Three milliliters of fermentation medium were collected from the test (5-azacytidine) and control (no 5-azacytidine) groups on days 2, 5, 8, 10, 12, 14, 16, and 18 of fermentation, and 6 mL of 70% ethanol was added. The medium was incubated in a water bath at 60 °C for 1 h, centrifuged at 4000 rpm for 15 min, and diluted accordingly. The red, yellow, and orange pigment values were measured using a spectrophotometer at wavelengths of 505 nm, 410 nm, and 465 nm, respectively, with absorbance values ranging from 0.2 to 0.8 (UV-2450, Shimadzu Corporation, Kyoto, Japan). The following quantitative formula was used: *Monascus* pigment value (U/mL) = absorbance × dilution factor.

### 2.4. Determination of Monacolin K Yield

Five milliliters of fermentation medium were collected from the test (5-azacytidine) and control (no 5-azacytidine) groups on days 2, 5, 8, 10, 12, 14, 16, and 18 of fermentation, and 15 mL of 70% methanol was added. The culture was disrupted using an ultrasonic crashing machine (SB25-120, Xinzhi Biological Polytron Technologies Company Inc., Nanjing, China) for 10 min, kept in the dark for 6 h, and analyzed by high-performance liquid chromatography (HPLC) (20A HPLC, Shimadzu Corporation, Kyoto, Japan) to determine the yields of Monacolin K. High performance liquid chromatography was performed using an Inertsil ODS-3 C18 column (150 mm × 4.6 mm × 5 mm). One milliliter of the supernatant was passed through a 0.22 μm organic filter membrane into a liquid vial, and the yield was determined by high performance liquid chromatography (HPLC). The mobile phase was methanol-phosphoric acid (3:1, *v*/*v*), operating at 1 mL/min. The eluate was monitored at 237 nm with ultraviolet spectroscopy (SPD-M20A, UV detector, Shimadzu Corporation, Kyoto, Japan). By diluting Lovastatin standard samples, a standard curve was determined and plotted (y = 30547x + 5135.9).

### 2.5. Scanning Electron Microscopy (SEM)

Two milliliters of fermentation medium were collected and centrifuged for 5 min at 12,000 rpm. *Monascus* M1 and RP2 cells, cultivated for 8 days, were resuspended in 2.5% glutaraldehyde solution for 12 h. The cells were washed twice with 0.1 M phosphate-buffered saline (PBS, pH = 7.4), and the supernatants were discarded. The cells were dehydrated using a series of ethanol (30%, 50%, 70%, 80%, 90%, and 100%), with each concentration washed for 10 min and centrifuged for 5 min at 12,000 rpm at 4 °C, followed by supernatant removal. Ethanol was replaced with isoamyl acetate and ethanol (*v*:*v* = 1:1) and an isopentyl acetate solution, with the cells resuspended in each solvent for 10 min and centrifuged under the same conditions. Hexamethyldiazabane was added to cover the cells, and the centrifuge tube was sealed with absorbent cotton and kept in an oven at 60 °C until the sample was dried [20,21]. The dried samples were observed using a scanning electron microscope (SU8010, Hitachi, Tokyo, Japan).

### 2.6. Statistical Analysis

All experiments were performed independently in triplicate. Statistical analysis was performed using SPSS Statistics 24 software. One-way analysis of variance (ANOVA) was used to assess the significance of differences between the control and experimental group results.

## 3. Results

### 3.1. Addition of 5-Azacytidine to Monascus M1 and RP2 at Different Time Points and Concentrations

5-azacytidine primarily affects the G1 phase of cell mitosis, although the specific period of the G1 phase in *Monascus* is unknown. Thus, we experimentally determined the best time for 5-azacytidine addition. The highest Monacolin K yield in *Monascus* M1 (Figure 1A) and the highest red pigment value in *Monascus* RP2 (Figure 1B) were observed when 5-azacytidine was added 24 h after *Monascus* was transferred from the seed fermentation medium to the main fermentation medium (*p* < 0.05). At this point, both *Monascus* M1 and RP2 had entered the G1 phase. As the concentration of 5-azacytidine increased, the red pigment value of *Monascus* RP2 (Figure 1C) and the Monacolin K yield of *Monascus* M1 (Figure 1D) also increased, reaching a peak at 10^−2^ g/L (*p* < 0.05). However, subsequent increases in concentration led to a gradual decrease in these values. This demonstrated that the methylation of 5-azacytidine had an effect on *Monascus* pigment production and Monacolin K yield.

When 5-azacytidine was added to *Monascus* cultures after 24 h, the red pigment value and Monacolin K level decreased to varying degrees. After 5 days of fermentation, secondary metabolite production in the experimental group exceeded that in the control group.

### 3.2. Effect of 5-Azacytidine on the Dry Weights of Monascus M1 and RP2

After 18 days of culture with 5-azacytidine, the concentration was adjusted to 10^−2^ g/L following an additional 24 h. Initially, 5-azacytidine slightly affected cell growth, and the dry weight of the experimental group (5-azacytidine) was lower than that of the control group on day 12 (no 5-azacytidine) (Figure 2A,B, *p* < 0.05). However, 5-azacytidine promoted the growth of *Monascus* RP2 (Figure 2A), initially increasing dry weight before inhibiting it. After 5-azacytidine addition, the dry weight of *Monascus* M1 and RP2 cells increased by 7.7% and 5.1% from 52 g/L to 56 g/L and from 41 g/L to 43.1 g/L, respectively. After 12 days, the dry weights reached the maximum of 56 g/L and 43.1 g/L, respectively. As fermentation progressed, the cells entered the decline phase, and dry weights gradually decreased.

### 3.3. Effect of 5-Azacytidine on Red, Orange, and Yellow Pigment Values of Monascus M1 and RP2

5-azacytidine enhanced the red pigment production of *Monascus* M1 (Figure 3A, *p* < 0.05) and RP2 (Figure 3B, *p* < 0.05). The red pigment value of *Monascus* M1 increased by 19.6%, from 52.3 U/mL to 62.2 U/mL, while that of *Monascus* RP2 increased by 15.6%, from 271.3 U/mL to 313.5 U/mL. On day 12, the red pigment value reached the maximum of 62.2 U/mL in *Monascus* M1 and 313.5 U/mL in *Monascus* RP2. After day 12, these values remained stable.

Notably, 5-azacytidine enhanced the orange pigment production of *Monascus* M1 (Figure 3C, *p* < 0.05) and RP2 (Figure 3D, *p* < 0.05). The orange pigment value of *Monascus* M1 increased by 16.3%, from 38.3 U/mL to 44.4 U/mL, while that of *Monascus* RP2 increased by 20.6%, from 151.4 U/mL to 181.5 U/mL. On day 12, the orange pigment value reached the maximum of 44.4 U/mL in *Monascus* M1 and 181.5 U/mL in *Monascus* RP2.

Additionally, 5-azacytidine enhanced the yellow pigment value of *Monascus* M1 (Figure 3E, *p* < 0.05) and RP2 (Figure 3F, *p* < 0.05). The yellow pigment value of *Monascus* M1 increased by 16.4%, going from 45.2 U/mL to 52.6 U/mL, while that of *Monascus* RP2 increased by 15.0%, from 201.3 U/mL to 231.5 U/mL. On day 12, the yellow pigment value reached the maximum of 52.6 U/mL in *Monascus* M1 and 231.5 U/mL in *Monascus* RP2.

### 3.4. Effect of 5-Azacytidine on the Monacolin K Yield of Monascus M1

5-azacytidine substantially increased Monacolin K yield in *Monascus* M1 (Figure 4, *p* < 0.05), with an increase of 58.6%, from 29.1 mg/L to 46.3 mg/L. The maximum yield of Monacolin K (46.3 mg/L) was achieved on day 12, after which the yield stabilized.

### 3.5. Effect of 5-Azacytidine on Mycelial Morphology of Monascus M1 and RP2

The surface of *Monascus* RP2 (Figure 5A–D) mycelium of the experimental group (Figure 5B,D) was more filamentous than that of the control group (Figure 5A,C). The spores of the control group were full and had a smooth surface, while those in the experimental group were rougher, with more wrinkles and protrusions. Similarly, the surface of *Monascus* M1 (Figure 5E–H) mycelium in the control group (Figure 5E,G) was smooth, with fewer filaments. The spores were full and showed no notable wrinkles. In contrast, the *Monascus* M1 mycelium in the experimental group (Figure 5F,H) had a rougher surface, with more filaments, and the spores were less full, displaying visible wrinkles.

SEM images showed that the hyphae in the control group maintained a complete, regular morphology. However, in the experimental group, the surface had more folds, ridges, and filaments.

## 4. Discussion

As a natural food colorant, *Monascus* pigment has been used in coloring sausage, ham, and other meat products [22]. *Monascus* pigments (MPs) are an azaphilone mixture of yellow, orange, and red pigments [23,24,25]. Some of the structural formulas of these pigments have been determined [26].

Many factors affect pigment production, and various strategies have been employed to improve yields [2]. The morphological change and enhanced pigment production of *Monascus* when co-cultured with *Saccharomyces cerevisiae* or *Aspergillus oryzae* have been studied, with medium composition adjustments made to increase pigment yields [27].

Chang et al. have suggested using response surface methodology to optimize the culture medium for Lovastatin production from *Monascus ruber*, providing a straightforward approach to improve Lovastatin production [28]. In addition, modifications to culture conditions, such as ventilation volume and stirring speed, have also been investigated [20,29]. These studies have explored various approaches to enhance *Monascus* secondary metabolite production. However, these methods primarily improve the growth and metabolism of Monaspergillus instead of *Monascus* by changing the exogenous environment and enhancing the ability of the cell to use carbon and nitrogen sources, thus providing limited theoretical support. Notably, 5-azacytidine, an effective regulator of DNA methylation, has been rarely studied in *Monascus*, making its effect on the fungus a topic of substantial scientific interest.

As research progresses, gene-editing technology has been increasingly applied to *Monascus*, and research on the regulatory mechanism of pigment production and Monacolin K in *Monascus* has advanced. Li et al. have identified *Mga1*, a G-protein alpha-subunit gene involved in regulating citrinin and pigment production in *Monascus* ruber M7 [30]. Jia et al. functionally characterized *MrbrlA* and *MrwetA* using knockdown and overexpression techniques. The deletion and overexpression of *MrbrlA* or *MrwetA* did not significantly alter conidial morphology, size, number, structure, or germination, but notably affected strain development and secondary metabolite production [31]. Liu et al. employed gene deletion, overexpression, and transcriptome analysis to investigate the function of *Monascus* M7 homologs, proposing that yeast protein transporters act as global regulators in fungi, significantly influencing pigment synthesis [32]. Xie et al. demonstrated that *pigR* is a key gene for *Monascus* pigment synthesis by overexpression and knockout, with its loss resulting in the inability of *Monascus* to produce pigments. Research on the effects of gene regulation on *Monascus* growth and metabolism has gradually improved [33]. However, studies on the effect of exogenous additions—such as carbon, nitrogen sources, and nutrients—remain limited, with few investigations into the methylation of *Monascus* itself.

As a potent DNA methylation-promoting agent, 5-azacytidine is widely used in epigenetic and cancer biology studies and for studying secondary metabolism in fungi. It does not provide energy for fungal growth and development but influences the synthesis of metabolites that are regulated by growth and secondary metabolism [34]. 5-azacytidine treatment enhanced the transcription of cellulase and xylanase genes in glucose-cultured humus strains and affected the development and secondary metabolism of *Aspergillus flavus* [35]. This is consistent with the results of the present experiment, in which the addition of 5-azacytidine promoted the production of *Monascus* pigment as well as Monacolin K. Therefore, it is of scientific interest to study the effects of 5-azacytidine on the synthesis of secondary metabolites of *Monascus*.

## 5. Conclusions

This study found a method to increase the production of secondary metabolites of the two strains, including Monacolin K and pigment in *Monascus* by adding 5-azacytidine either at the beginning of liquid fermentation of *Monascus* or during the fermentation process. Among them, Monacolin K on *Monascus* M1 and pigment on *Monascus* RP2 had the most significant effects. Electron microscope observation showed that strains with 5-azacytidine added had more wrinkles and filaments. We hypothesize that 5-azacytidine contributes to the generation of DNA methylation and that the DNA methylation of filamentous fungi can promote the increase in its metabolites.

We hypothesize that 5-azacytidine may also change other secondary metabolites of *Monascus* and those of other filamentous fungi especially citrinin production; however, future research is required to validate this hypothesis.

## Figures and Tables

**Figure 1 jof-10-00819-f001:**
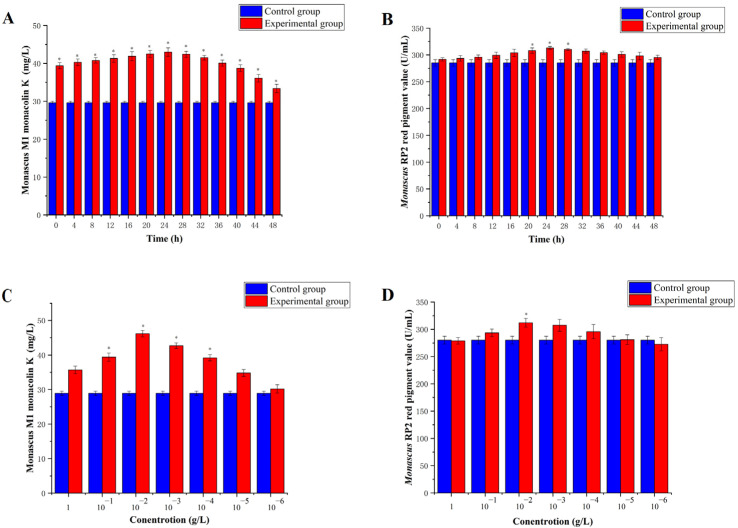
Effects of 5-azacytidine on *Monascus* M1 and *Monascus* RP2. (**A**) Yield of Monacolin K of *Monascus* M1 at different addition times. (**B**) Red pigment value of *Monascus* RP2 at different addition times. (**C**) Yield of Monacolin K of *Monascus* M1 at different concentrations of 5-azacytidine. (**D**) Red pigment value of *Monascus* RP2 at different concentrations of 5-azacytidine, * *p* < 0.05.

**Figure 2 jof-10-00819-f002:**
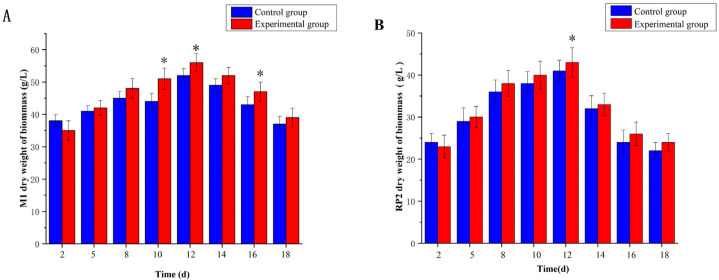
Effects of 5-azacytidine on *Monascus* M1 and *Monascus* RP2. (**A**) Dry weight of *Monascus* M1. (**B**) Dry weight of *Monascus* RP2, * *p* < 0.05.

**Figure 3 jof-10-00819-f003:**
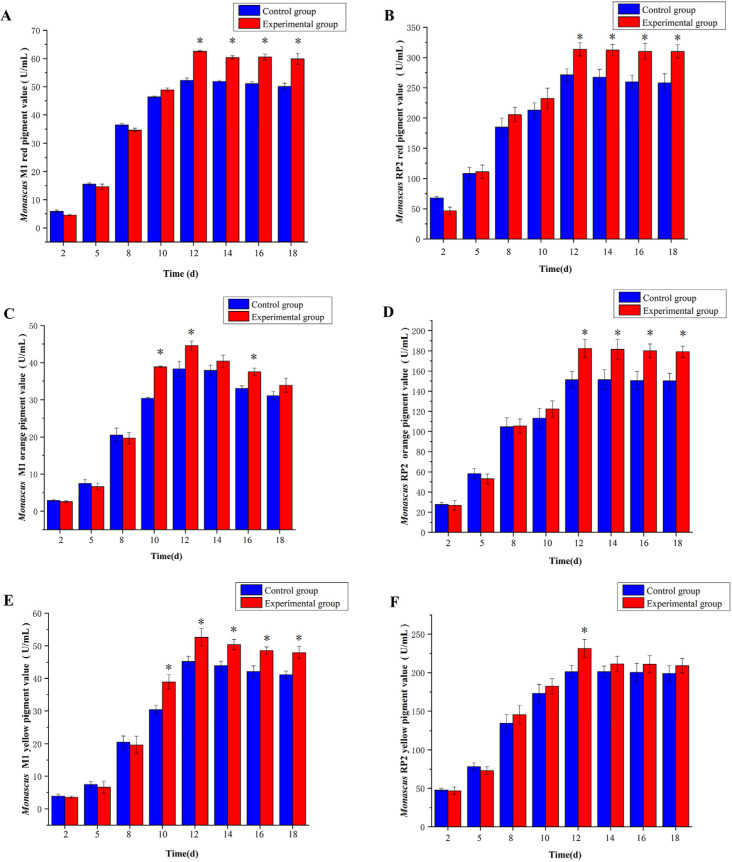
Effects of 5-azacytidine on the pigment value in *Monascus* M1 and *Monascus* RP2. (**A**) The red pigment value of *Monascus* M1, (**B**) the red pigment value of *Monascus* RP2, (**C**) the orange pigment value of *Monascus* M1, (**D**) the orange pigment value of *Monascus* RP2, (**E**) the yellow pigment value of *Monascus* M1, and (**F**) the yellow pigment value of *Monascus* RP2, * *p* < 0.05.

**Figure 4 jof-10-00819-f004:**
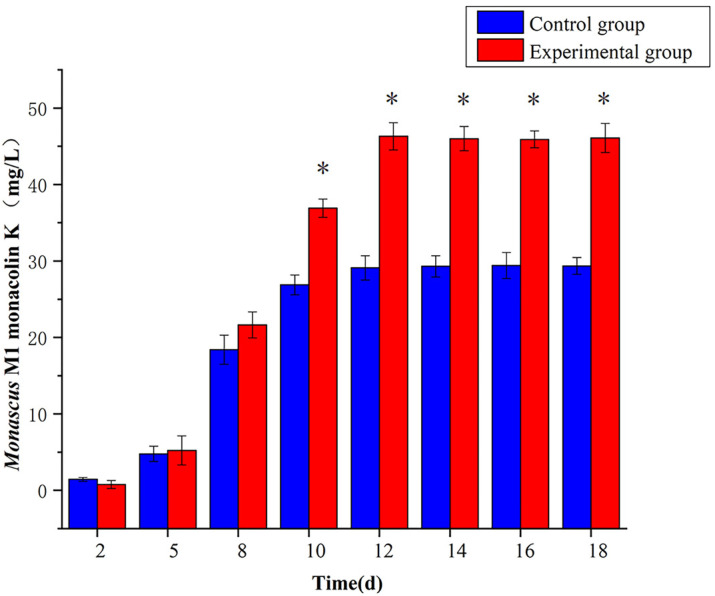
Yield of Monacolin K of *Monascus* M1, * *p* < 0.05.

**Figure 5 jof-10-00819-f005:**
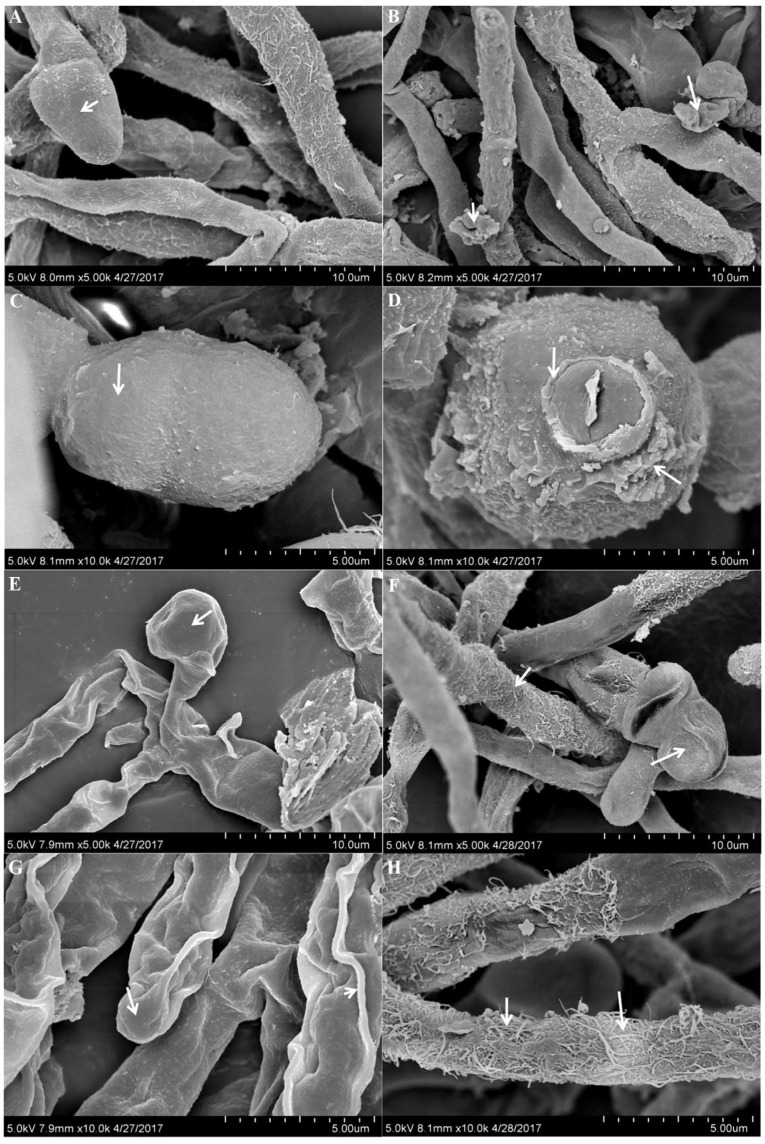
Scanning electron micrographs showing the morphology of *Monascus* RP2 (**A**–**D**) and *Monascus* M1 (**E**–**H**) at 8 days in different culture media with different magnification factors (5000× and 10,000×, respectively). *Monascus* RP2 (**A**,**C**) in the original medium, *Monascus* M1 (**E**,**G**) in the original medium, *Monascus* RP2 (**B**,**D**) in the medium containing 5-azacytidine, and *Monascus* M1 (**F**,**H**) in the medium containing 5-azacytidine. Arrows indicate wrinkles.

## Data Availability

Data are contained within this article.

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
