# Peer review of "Improvement of Monacolin K and Pigment Production in Monascus by 5-Azacytidine"

_jof, 2024, doi:10.3390/jof10120819_

Round 1
Reviewer 1 Report
The manuscript titled "Improvement of Monacolin K and pigment production in Monascus by 5-azacytidine" describes the effects that a DNA methylating agent produces on two Monascus strains, M1 and RP2, focusing on the values of red, yellow, and orange pigments as well as the monacolin K. In addition, to verify whether 5-azacytidine methylation affected it, an SEM study on hyphae morphology was included.
It should be interesting to know which pigments are involved.
The Introduction looks incomplete. Some relevant papers on the topic should be included (Example: doi 10.3389/fchem.2019.00763)
Results
3.1. Addition of 5-azacytidine to Monascus M1 and RP2 at different time points and concentrations
The comparison must include the control, accompanied by a statistical analysis. Please include it in Fig 1.
3.2. Effect of 5-azacytidine on the dry weights of Monascus M1 and RP2
Before (or in addition to) testing the different concentrations of 5-azacytidine, the MIC (minimal inhibitory concentration) should be determined.
Figure 2: Is there any statistical difference between the control versus the treated group?
3.3. Effect of 5-azacytidine on red, orange, and yellow pigment values of Monascus 183 M1 and RP2.
In the experimental section, please define the U/mL you use to analyze the pigment concentration.
Fig 3: are the control and experimental groups statistically different?
3.4. Effect of 5-azacytidine on the Monacolin K yield of Monascus M1
Include the column you used in the HPLC method and the calibration curve to determine the Monacolin K concentration. Include statistics.
3.5. Effect of 5-azacytidine on mycelial morphology of Monascus M1 and RP2.
Please re-analyze the results obtained and rewrite this section. Adding 5-Azacytidine brings malformations according to the concentration applied next to the MIC value. It is more complex than exposing more surface to permeate secondary metabolites.
Complete the legend of Fig. 5
--
Author Response
Comments 1: Addition of 5-azacytidine to Monascus M1 and RP2 at different time points and concentrations.
The comparison must include the control, accompanied by a statistical analysis. Please include it in Fig 1.
Response:Thank you for your suggestion.We have greatly revised the full text according to your suggestions.
Data were analyzed by One-way analysis of variance (ANOVA) within groups. In this experiment, in order to explore the best addition concentration and addition time, the difference between adjacent groups was not significant, but the difference of a single concentration within the whole group was significant. The analysis results were significant at different time and different concentrations (p<0.05).
Comments 2: Effect of 5-azacytidine on the dry weights of Monascus M1 and RP2
Before (or in addition to) testing the different concentrations of 5-azacytidine, the MIC (minimal inhibitory concentration) should be determined.
Response:Thank you for your suggestion.In this experiment, based on the optimal concentration obtained in Figure 1, the effect of 5-azacytidine on Monascus biomass under the optimal condition was determined, and the difference between the groups was significant (p<0.05).
Comments 3: Effect of 5-azacytidine on red, orange, and yellow pigment values of Monascus 183 M1 and RP2.
In the experimental section, please define the U/mL you use to analyze the pigment concentration.
Fig 3: are the control and experimental groups statistically different?
Response:Thank you for your suggestion.We have supplemented the unit definition section as well as the significance analysis in Figure 3.
Comments4: Effect of 5-azacytidine on the Monacolin K yield of Monascus M1
Include the column you used in the HPLC method and the calibration curve to determine the Monacolin K concentration. Include statistics.
Response:Thank you for your suggestion.We have added Monacolin K to the article (y = 30547x + 5135.9 R2 = 0.9992) and performed statistical analysis of the data.
Comments 5: Effect of 5-azacytidine on mycelial morphology of Monascus M1 and RP2.
Please re-analyze the results obtained and rewrite this section. Adding 5-Azacytidine brings malformations according to the concentration applied next to the MIC value. It is more complex than exposing more surface to permeate secondary metabolites.
Complete the legend of Fig. 5
Response:Thank you for your suggestion.The purpose of this study was to explore the optimal concentration of 5-azacytidine to promote the growth and metabolism of Monascus. It was found that 5-azacytidine had a certain effect on the morphology of Monascus, and the secondary metabolites also had corresponding changes. In the future, we will consider this issue and study it in depth.

Reviewer 2 Report
As mentioned above, I fail to see the significance of the work done so far. The experiments were nicely perfomed, but lack description (see methods above) and rigorous evaluation (see results and discussion). I thus doubt the scientific significance of the present work. For details, please see the sections above.
See above
Author Response
- The introduction is insufficient right now. The first paragraph is all right, but the second is a mess and I did not manage to follow it. The first couple of paragraphs in the discussion should be in the introduction instead and give a sufficient idea of what is known about the pigments and Monaculin A. I also miss some information on the differences and similarities of the two used Monascus strains and the reasons why those were chosen. The aim of the whole manuscript is also not sufficiently explained.
Response:Thank you for your suggestion. We have revised the discussion and introduction section to remove unnecessary and wrong parts. For the interpretation of strain information, the main purpose was to explore the effect of 5-azacytidine on the important metabolite pigment and Monacolin K of Monacolin K. Representative strains with high Monacolin K and high pigment production were selected.We have also revised these into the article.
- The method section is missing vital information and unclear in parts. - What does 'preserved in our laboratory (l. 66) mean? - Why were different media used for the two strains? - In the results, the optimal time to add 5-azacytidine is supposed to be after 24 h (l. 159), in the methods, 48 h are suddenly optimal (ll. 86 f.) - Ll. 87-89 indicate that 5-azacytidine was added slowly over some time until a final concentration of 10^-3 g/L was reached. In the subsequent sentence, it was suddenly added at a single time point. What is correct? - Everywhere: What is pigment value? Should it not be pigment concentration? And why is it given in Units if it is not an enzyme? - What is 'diluted corresponding multiple' (l. 97). - How was the retention time of Monascolin K determined? No standard is mentioned in the HPLC paragraph nor in the results. - Was the sample not centrifuged or filtered before injecting it into the HPLC? - What was the pH of the PBS (l. 116)? - When was the fermentation broth for SEM collected? - What is the composition and concentration of isopentyl acetate (l. 120)? - How much hexamethyldisilazane was added (l. 122)? - The statistical analyses are nicely described here, but never mentioned in the actual result section - and sorely missing there!
Response:Thank you for your suggestion.
The major secondary metabolite was Monacolin K in M1 strain and pigment in RP2 strain. The optimal medium for different strains is different, and the appropriate medium is more conducive to the production of secondary metabolites. The optimal adding time of 5-azacytidine was incorrectly described in the experimental method. The optimal concentration and optimal time of 5-azacytidine were re-described.
The color value was determined by ultraviolet spectrophotometer. The following quantitative formula was used: Monascus pigment value (U/mL) = absorbance × dilution factor. The method and retention time of Monascolin K were from the literature method. The mother liquor was prepared with lovastatin standard, and the standard curve was determined by gradient dilution. The sample was processed as follows :2 mL of fermentation broth was placed in a centrifuge tube, 6 mL of 75% methanol solution (v/v, 75:25) was added, sonicated for 30 min at room temperature, and left to be stored in the dark overnight. One mL of the supernatant was passed through a 0.22 μm organic filter membrane into a Liquid vial, and the yield was determined by High Performance Liquid Chromatography (HPLC). It is now added to the text. Significance analysis has also been supplemented to the text.
Hexamethyldisilazane is appropriate, and no bacteria can be used, so there is no specific dosage, which has been updated in the text
- The results are all right for the most part, but can be drastically improved. One of the most important point is the significance of the described differences. Where are the tests? Especially for the data in Fig 1 and 2, I can hardly believe those data points to be significantly different. The text obviously has to be adjusted according to the significance testing results. In addition, the discussion is severly lacking right now. The first couple of paragraphs should be part of the introduction (and no, Monascus is not red grains or red rice, it is just what makes the grain/rice red! (l. 366)). There is no mention of the actual results in the discussion and no combination/comparison of those with the literture. Right now, the discussion is pretty much non-existant. And the last sentence is a pure exaggeration (l. 394) since they only tested a couple of pigments and Monasculin and can thus by no way claim that the managed to increase 'most of the secondary metabolites'. - ll. 137 and 140: What does 'transferred to the fermentation broth from the liquid medium' mean? I do not understand based on the method section. - l. 140: The fact that the strains reached the G1 period is sorely a hypothesis at this time point as it has not been proven by anything but the slight increase (is that even significant?) in Monasculin K and pigment production. It should be portrayed as such. - l. 141 and 142: RP2 is Fig 1 D and M1 is Fig 1C. - l. 146: 'methylation of 5-azacytidine': First of all, not azacytidine is methylated, but it is supposed to methylate the DNA. Second, this is not proven here as a matter of fact, but another hypothesis based on the results. - Where is the dataset going with the paragraph from ll. 148 to 151? - section 3.3: What time point are these measurements from? - Please adjust the y-axes in Fig. 3 to be identical for all three pigments with the same strain to make it easier for the reader to see differences in concentration levels. - ll. 263-269: This belongs in the introduction. It also fails to convince me why the production of Monacolin K has to be improved. Just because of similarities to the synthesis of lovastatin? How high are these similarities? Do the authors want to imply that Monascolin has similar health benefits? If so, they do not say. - l. 273: No HPLC is shown, but an HPLC chromatogram. Which, by the way, is not really needed here and can go into the supplement. It also does not need every single minor peak to be annotated with the peak area, that just looks messy. - Why is Fig. 4a so tiny? - What is the difference between Fig 5 A and C etc.? The zoom level? Then please say so in the respective legend. I also think that this figure, and possibly also Fig. 3 would benefit from legends outside the figure indicating what is in which line and column to make it easier to grasp the info without refering to the figure legend. - The start of the figure legend of legend 5 is missing.
Response:Thank you for your suggestion.
In Figure 1 and Figure 2, there are some direct differences between different groups, but the experiment found that the 5-azacytidine has a limited effect on the pigment, but a significant effect on mk. The discussion section has been rewritten to remove inaccurate portions of the text.
The last word is our conjecture, because 5-azacytidine affects the production of pigment and mk, the main products of monascus, and has a significant effect on the growth and appearance of the bacteria. Therefore, it is speculated that it also has an effect on other metabolism, which will be supplemented in future experiments. The inaccurate description has been corrected. The G1 phase of fungi is mainly in the early fermentation period, so it is just satisfied within 48h of the preparation time of seed liquid. Therefore, experiments were carried out, and it was found that the effect of 5-azacytidine on the bacteria did indeed peak.
The screening of the optimal concentration and time has been changed, and the question about the unclear legend has been changed. Monacolin K, as a kind of lovastatin, is beneficial to improve the symptoms of hyperlipidemia and hypertension, and has become one of the ideal drugs recognized by the medical community at present. Monacolin K exists in both acid and lactone forms. The lactone form needs to be converted to the acid form by the hydrolysis of carboxyesterase in humans. However, the structure of monascus natural fermentation is mostly acid form, so it can directly play a lipid-lowering role without conversion. Therefore, the production improvement of Monacolin K makes sense. HPLC chromatogram mainly showed the peak time of mk and the difference of area between groups, and the peak position of Monacolin K has also been clearly marked. Figure 5 A and C are indicated in the legend for different scaling levels, and their scaling levels are shown at the bottom of the figure.
Reviewer 3 Report
The manuscript written by Zhang et al. explored the impact of 5-azacytidine addition on the metabolite production of Monascus M1 and RP2. Through evaluating the changes of dry weight, pigment values, monacolin K yield and hyphae, the 5-azacytidine treatment can indeed result in improvements on metabolite production.
The goal and design of this work are clear. However, the introduction and discussion parts need to be more organized.
1. The introduction section is not abundant. For example, in the line 36, what does ctnA specifically stand for? It is better to give a detailed explanation for citrinin example. In the line 51, please list some examples of metabolism regulation of 5-azacytidine as the DNA methylator and reasons that it might be effective on Monascus.
2. In introduction section, it is recommended to display the chemical structures of 5-azacytidine, monacolin K, citrinin and other related secondary metabolites regulated by 5-azacytidine in a form of figure.
3. As for the method of determination of pigment value, three milliliters of broth were collected every two days in a row. How did it keep the measurement accurate with the consuming of broth? Please explain for this.
4. For Figure 4, it is recommended to supplement the authentic monacolin K as the control only when comparing by HPLC.
5. The discussion section is not well-organized and insufficient. For example, in the line 374, the components of pigments can be discussed a little more. In the line 377, the literatures on co-cultivation between Monascus and other microorganisms are also supposed to discuss. Additionally, the related paragraphs can be combined.
6. Based on results this work obtained, it is worth to discuss how to dig up the activated secondary metabolites by 5-azacytidine in future studies.

Author Response
Comments 1: The introduction section is not abundant. For example, in the line 36, what does ctnA specifically stand for? It is better to give a detailed explanation for citrinin example. In the line 51, please list some examples of metabolism regulation of 5-azacytidine as the DNA methylator and reasons that it might be effective on Monascus.
Response:Thank you for your suggestion. What was not clear about the introduction section has been changed to add an example of 5-azacytidine as a DNA methylating agent to regulate metabolism and why
Comments 2: In introduction section, it is recommended to display the chemical structures of 5-azacytidine, monacolin K, citrinin and other related secondary metabolites regulated by 5-azacytidine in a form of figure.
Response:Thank you for your suggestion.The mechanism of 5-azacytidine in the regulation of Monascus is still unclear. In this study, the effects of 5-azacytidine on secondary metabolites and the growth appearance of Monascus were investigated.The mechanism will be further studied in the following experiments.
Comments 3: As for the method of determination of pigment value, three milliliters of broth were collected every two days in a row. How did it keep the measurement accurate with the consuming of broth? Please explain for this.
Response:Thank you for your suggestion.For sampling, 3mL of the original fermentation broth was sucked with a pipetting gun, and the same specifications of the nozzle and pipetting gun were used for each sampling to ensure the accuracy of each sampling. The samples were processed similarly using the same reagents in the same environment, and the assay methods were kept consistent.
Comments 4: For Figure 4, it is recommended to supplement the authentic monacolin K as the control only when comparing by HPLC.
Response:Thank you for your suggestion.In order to illustrate the difference of Monacolin K content between different groups, the peak time of the standard material was consistent with the marked position in the figure, so it was not added.
Comments 5: The discussion section is not well-organized and insufficient. For example, in the line 374, the components of pigments can be discussed a little more. In the line 377, the literatures on co-cultivation between Monascus and other microorganisms are also supposed to discuss. Additionally, the related paragraphs can be combined.
Response:Thank you for your suggestion.We fully adopted your comments, supplemented the insufficient discussion, fully discussed the scientific significance of 5-azacytidine and the relevance of the article with it, and combined the paragraphs
Comments 6: Based on results this work obtained, it is worth to discuss how to dig up the activated secondary metabolites by 5-azacytidine in future studies.
Response:Thank you for your suggestion.We discussed how to explore the effect of 5-azacytidine on Monascus.

Round 2
Reviewer 1 Report
Comments:
Section: Addition of 5-azacytidine to Monascus M1 and RP2 at different time points and concentrations.
The comparison must include the control, accompanied by a statistical analysis. Please include it in Fig 1.
Also, add in Fig. 1 (C) the incubation time when you measured the yield of monacolin K of Monascus M1 at different concentrations of 5-azacytidine plus the control at that time. Also, add the incubation time when you measured and the control in Fig. 1 (D)
Section: Effect of 5-azacytidine on the Monacolin K yield of Monascus M1
Include the column you used in the HPLC method
Section: Effect of 5-azacytidine on mycelial morphology of Monascus M1 and RP2.
Delete the phrase: “We hypothesize that these increased folds, protrusions, and filaments increase the cell surface area, facilitating the secretion of secondary metabolites by Monascus.” The experiments performed and discussed in this manuscript do not add evidence to formulate this hypothesis.
Revise the phrase: Chang et al., have suggested using response surface methodology to optimize the culture medium for lovastatin production from Monascus ruber, providing a straightforward approach to improve lovastatin production [28]
Please check the use of italics when writing scientific names.
A major drawback of Monascus pigments for food is the possibility of containing citrinin. It would be interesting to consider in the general discussion and perhaps for the future of this research, the study of how the expression of the citrinin pathway is modified under the effect of 5-azacytidine.
-
Author Response
Comments 1: Addition of 5-azacytidine to Monascus M1 and RP2 at different time points and concentrations.
The comparison must include the control, accompanied by a statistical analysis. Please include it in Fig 1.
Also, add in Fig. 1 (C) the incubation time when you measured the yield of monacolin K of Monascus M1 at different concentrations of 5-azacytidine plus the control at that time. Also, add the incubation time when you measured and the control in Fig. 1 (D).
Response 1: Thank you for your suggestion.We have revised Figure 1 to include a control group and a significance analysis.
Comments 2: Section: Effect of 5-azacytidine on the Monacolin K yield of Monascus M1 Include the column you used in the HPLC method.
Response 2: Thank you for your suggestion.The chromatographic column used in the HPLC method is C18, which has now been added in the article.(High performance liquid chromatography was performed using an Inertsil ODS-3 C18 column (150 mm×4.6 mm×5 mm)).
Comments 3.Section: Effect of 5-azacytidine on mycelial morphology of Monascus M1 and RP2. Delete the phrase: “We hypothesize that these increased folds, protrusions, and filaments increase the cell surface area, facilitating the secretion of secondary metabolites by Monascus.” The experiments performed and discussed in this manuscript do not add evidence to formulate this hypothesis.
Response 3: Thank you for your suggestion.We removed the phrase.
Comments 4. Revise the phrase: Chang et al., have suggested using response surface methodology to optimize the culture medium for lovastatin production from Monascus ruber, providing a straightforward approach to improve lovastatin production [28].
A major drawback of Monascus pigments for food is the possibility of containing citrinin. It would be interesting to consider in the general discussion and perhaps for the future of this research, the study of how the expression of the citrinin pathway is modified under the effect of 5-azacytidine.
Response 4: Thank you for your suggestion.Text writing issues have been corrected and highlighted in the original text. The first round of modifications is marked in yellow, and the second round of modifications is marked in blue.

Reviewer 2 Report
A lot of my original comments remained unanswered and/or did not lead to any changes in the manuscript. In addition, smaller changes in the manuscript were not marked, which is extremely annoying as it costs more time during re-reviewing.
I think that the manuscript was improved through the revision, but not enough so to enable publishing. See above for the reasons.
Please recheck my original comments from the first revision.
If you plan to publish this, please get rid of the 'pigment value' and chose a correct scientific term.
In the abstract, through (unmarked) changes of the text, it is now not possible to assign fungal strain and pigment to the percentages given.
Please make sure that all strain names are given in italics.
The legend of Figure 1 ended up somewhere far below the figure itself.
The sentence '5-azacytidine may have broad applications' (end of section 3.2) belongs in the discussion, not here.
Author Response
Comments 1: What is 'monascus pigment value' (end of section 2.3). Why is it given in U if not an enzyme? This does not make sense at all and neither did the answer I received last time. You cannot use a different standard than the actual molecule you are trying to measure! If you do so, you may semi-quantify, but not quantify.
Are the results presented clearly and in sufficient detail, are the conclusions supported by the results and are they put into context within the existing literature?
Response1: Thank you for your suggestion. In the study of Monascus, the use of U to represent the pigment value is widely used. Many of them are reflected in the article. Such as(【1】Li, W., Li, Y., Yu, W., Li, A., & Wang, Y. (2022). Study on production of yellow pigment from potato fermented by Monascus. Food Bioscience, 50, 102088. https://doi.org/10.1016/j.fbio.2022.102088. 【2】Yang, S., Zhou, H., Dai, W., Xiong, J., & Chen, F. (2021). Effect of Static Magnetic Field on Monascus ruber M7 Based on Transcriptome Analysis. Journal of Fungi, 7(4), 256. https://doi.org/10.3390/jof7040256. 【3】 Zhang, S., Shu, M., Gong, Z., Liu, X., Zhang, C., Liang, Y., Lin, Q., Zhou, B., Guo, T., & Liu, J. (2024). Enhancing extracellular monascus pigment production in submerged fermentation with engineered microbial consortia. Food Microbiology, 121, 104499. https://doi.org/10.1016/j.fm.2024.104499).
Comments 2: Many of my comments in the original review were not answered nor incorporated into the text for section 3.1 to 3.3. The significance test results were added to the figures (none in fig 1 - not done or no significant results?), but not implemented in changes of the text. Information on Monacolin K (start of section 3.4) still belongs in the introduction, not the results. The discussion has been thoroughly improved and now presents a lot of surrounding info in an understandable way, but fails to combine the presented data from the manuscript with the information from literature - the main job of a discussion. The first paragraph of the conclusion still (!) claims that '5-azacytidine increased the secondary metabolites of the two strains, including....' naming all tested metabolites. It thus wrongly indicates that further metabolites were investigated.
Response2: We apologize for the previous response, as the entire text has been modified, we have highlighted the main changes in yellow for ease of viewing. On the basis of the last modification, we have used blue highlighting to emphasize and highlight. In view of the number of problems you have raised and our professional perspective, we actively absorb opinions and correct them. In order to show our respect for you, we will list each question separately for your approval, as follows:
Comments (1): What is 'monascus pigment value' (end of section 2.3). Why is it given in U if not an enzyme?
Response (1): In the study of Monascus, the use of U to represent the pigment value is widely used. Many of them are reflected in the article.(【1】【2】【3】).
Comments (2): In the abstract, through (unmarked) changes of the text, it is now not possible to assign fungal strain and pigment to the percentages given.
Response (2):The pigment percentage in the abstract indicates that it has been corrected and has been highlighted.
Comments (3): Please make sure that all strain names are given in italics.
Response (3): It's been checked out.
Comments (4):The legend of Figure 1 ended up somewhere far below the figure itself.
Response (4): The figures and legends have been changed, perhaps due to the format software format caused a problem version problem.
Comments (5): The sentence '5-azacytidine may have broad applications' (end of section 3.2) belongs in the discussion, not here.
Response (5): (end of section 3.2) It has been modified and highlighted.
Comments (6): I also miss some information on the differences and similarities of the two used Monascus strains and the reasons why those were chosen. The aim of the whole manuscript is also not sufficiently explained.
Response (6): For the interpretation of strain information, the main purpose was to explore the effect of 5-azacytidine on the important metabolite pigment and Monacolin K of Monacolin K. Representative strains with high Monacolin K and high pigment production were selected.We have also revised these into the article.
Comments (7): In the results, the optimal time to add 5-azacytidine is supposed to be after 24 h (l. 159), in the methods, 48 h are suddenly optimal (ll. 86 f.) - Ll. 87-89 indicate that 5-azacytidine was added slowly over some time until a final concentration of 10^-3 g/L was reached. In the subsequent sentence, it was suddenly added at a single time point.
Response (7): The optimal adding time of 5-azacytidine was incorrectly described in the experimental method. The optimal concentration and optimal time of 5-azacytidine were re-described.
Comments (8): Why were different media used for the two strains?
Response (8): The major secondary metabolite was Monacolin K in M1 strain and pigment in RP2 strain. The optimal medium for different strains is different, and the appropriate medium is more conducive to the production of secondary metabolites.
Comments (9): How was the retention time of Monascolin K determined? No standard is mentioned in the HPLC paragraph nor in the results.
Response (9): We determined its standard curve and determined the retention time through the peak time of the standard curve, and drew a standard curve, which has been added to the article.
Comments (10): Was the sample not centrifuged or filtered before injecting it into the HPLC?
Response (10): One mL of the supernatant was passed through a 0.22 μm organic filter membrane into a Liquid vial, and the yield was determined by High Performance Liquid Chromatography (HPLC). It is now added to the text. Significance analysis has also been supplemented to the text.
Comments (11): What was the pH of the PBS (l. 116)? - When was the fermentation broth for SEM collected?
Response (11): The ph of PBS used in the experiments was 7.4.The fermentation broth was collected at 2,5,8,12,15, and 18d of fermentation.
Comments (12): What is the composition and concentration of isopentyl acetate (l. 120)? - How much hexamethyldisilazane was added (l. 122)?
Response (12): Hexamethyldisilazane is appropriate, and no bacteria can be used, so there is no specific dosage, which has been updated in the text。Isopentyl acetate was pure and not diluted.
Comments (13): The text obviously has to be adjusted according to the significance testing results.
Response (13): The significance analysis of the data has been recalculated and added to the text.
Comments (14): What does 'transferred to the fermentation broth from the liquid medium' mean?
Response (14): Description errors have been corrected and highlighted in the article.
Comments (15): I do not understand based on the method section. - l. 140: The fact that the strains reached the G1 period is sorely a hypothesis at this time point as it has not been proven by anything but the slight increase (is that even significant?) in Monasculin K and pigment production.
Response (15): The G1 phase of fungi is mainly in the early fermentation period, so it is just satisfied within 48h of the preparation time of seed liquid. Therefore, experiments were carried out, and it was found that the effect of 5-azacytidine on the bacteria did indeed peak.
Comments (16): section 3.3: What time point are these measurements from? - Please adjust the y-axes in Fig. 3 to be identical for all three pigments with the same strain to make it easier for the reader to see differences in concentration levels.
Response (16): We have thought deeply about your opinion. The measured data show that there are three pigments of red, orange and yellow of two strains, and the combined treatment effect is not very good, so the original legend is retained.
Comments (17): This belongs in the introduction. It also fails to convince me why the production of Monacolin K has to be improved. Just because of similarities to the synthesis of lovastatin? How high are these similarities? Do the authors want to imply that Monascolin has similar health benefits? If so, they do not say.
Response (17): The reason for the production of Monacolin K is reflected in the introduction and discussion in the article, which is highlighted and marked.
Comments (18): l. 273: No HPLC is shown, but an HPLC chromatogram. Which, by the way, is not really needed here and can go into the supplement. It also does not need every single minor peak to be annotated with the peak area, that just looks messy. - Why is Fig. 4a so tiny? - What is the difference between Fig 5 A and C etc.?
Response (18): Figure 5 The difference between A and C is that Monacolin K is presented in different ways. However, we cannot modify the rest of the peak value due to the computer data update, and only this figure is used as a display. We apologize for this.

Reviewer 3 Report
The revised manuscript by Zhang et al. underscores the potential of 5-azacytidine and offers deeper insights into its impact on metabolite production. Furthermore, the study expands the interest in the role of 5-azacytidine in the biosynthesis of secondary metabolites in Monascus.
No further comments.

Author Response
Comments 1: The revised manuscript by Zhang et al. underscores the potential of 5-azacytidine and offers deeper insights into its impact on metabolite production. Furthermore, the study expands the interest in the role of 5-azacytidine in the biosynthesis of secondary metabolites in Monascus.
Response 1: Thank you for your review and suggestions, which are very important for improving the quality of our manuscript.

Round 3
Reviewer 1 Report
From my perspective, this manuscript could be acceptable after making minor revisions. Below are my detailed observations:
- General: Please review the punctuation and the use of italics for scientific names.
- Discussion. In the last paragraph, the term "conclusion products" is mentioned. Could you please clarify or rephrase this term to ensure that it is more accurately defined or explained?
- Conclusions. Please remove the following sentence:
"Simultaneously, the folds and filaments increase the surface area of the cell and promote metabolite excretion from the body, thereby increasing the amount of secondary metabolites."
There is no experimental evidence to support this statement, so it cannot be concluded at this stage.
From my perspective, this manuscript could be acceptable after making minor revisions. Below are my detailed observations:
- General: Please review the punctuation and the use of italics for scientific names.
- Discussion. In the last paragraph, the term "conclusion products" is mentioned. Could you please clarify or rephrase this term to ensure that it is more accurately defined or explained?
- Conclusions. Please remove the following sentence:
"Simultaneously, the folds and filaments increase the surface area of the cell and promote metabolite excretion from the body, thereby increasing the amount of secondary metabolites."
There is no experimental evidence to support this statement, so it cannot be concluded at this stage.
Author Response
Comments 1: General: Please review the punctuation and the use of italics for scientific names.
Response1:Thank you for your suggestion. We have thoroughly checked the manuscript from beginning to end and corrected any errors.
Comments 2: Discussion. In the last paragraph, the term "conclusion products" is mentioned. Could you please clarify or rephrase this term to ensure that it is more accurately defined or explained?
Response2: Thank you for your suggestion. We believe that the term "conclusion products" is inaccurate and have changed it to "metabolite".
Comments 3: 3.Conclusions. Please remove the following sentence:
"Simultaneously, the folds and filaments increase the surface area of the cell and promote metabolite excretion from the body, thereby increasing the amount of secondary metabolites."
There is no experimental evidence to support this statement, so it cannot be concluded at this stage.
Response3: Thank you for your suggestion. To avoid ambiguity, we have removed this speculative conclusion.
Reviewer 2 Report
See the detailed comments above
Some strain names are still not in italics, e.g. in the discussion, second paragraph Aspercillus oryzae and Saccharomyces cerevisiae.
Author Response
Point-by-point responses to the reviewers' comments:
Journal of Fungi
We would like to thank the reviewers for their constructive and helpful comments. We have revised the manuscript accordingly, as described below.
Comments 1:You used lovastatin as standard to quantify monacolin K yield. That does not work. You either have to use the exact same molecule (so monacolin K) as standard, or you are only allowed to call what you do semi-quantification.
Response1:Thank you for your suggestion. The structure of lovastatin and monacolin K is completely consistent, which has been confirmed by relevant literature. In the field of medicine, it is called lovastatin, and it is commonly expressed as monacolin K in the research field of Monascus. Such as(【1】Xiong, Z.; Cao, X.; Wen, Q.; Chen, Z.; Cheng, Z.; Huang, X.; Zhang, Y.; Long, C.; Zhang, Y.; Huang, Z. An Overview of the Bioactivity of Monacolin K / Lovastatin. Food and Chemical Toxicology 2019, 131, 110585, doi:10.1016/j.fct.2019.110585.【2】Wang, J.; Liang, J.; Chen, L.; Zhang, W.; Kong, L.; Peng, C.; Su, C.; Tang, Y.; Deng, Z.; Wang, Z. Structural Basis for the Biosynthesis of Lovastatin. Nat Commun 2021, 12, 867, doi:10.1038/s41467-021-21174-8)
Comments 2: The labelling of the x-axis in Figure 1 D was lost during revision. - The text claims that cell growth and dry weight of the experimental group was lower than that of the control group initially (section 3.2) and gives a p-value below 0.05. But there is no indication for a significant difference between both values in the respective bar at d 2 (or 4 or...) of either figure 2 A or B.
Response2: Thank you for your suggestion.The problem with the abscissa in Figure 1 (d) has been modified. As for Figure 2, the biomass problem was clearly located on the twelfth day.The biomass discussed in this article is also a reference for the change of bacterial morphology by 5-azacytidine. These results suggest that 5-azacytidine May affect cell morphology and lead to metabolic changes.
Comments 3: I also find that the differences induced by the addition of 5-azacytidine are on most days not significant and very slight. This should be reflected in the text, but is not. - Section 3.3: The values of orange pigment do not remain stable at all for Monascus M1. Please correct the text accordingly.
Response3:Thank you for your suggestion.We add to the article the limitations of the effect of 5-azacytidine on biomass.Inaccurate descriptions of orange and yellow pigments have been changed and removed. (After day 12, these values remained stable.)
Comments 4: Fig. 4 B and C is really not helpful at all to the reader. It can go into the supplement or be deleted from the manuscript completely. It is also not an HPLC profile, but a chromatogram...
Response4:Thank you for your suggestion.The chromatogram was added to prove the reliability of the data, and to let readers intuitively feel the increase of monacolin K and the difference of monacolin K between different groups.
Comments 5: I also feel that the part on genetic modification of Monascus in the discussion is quite long for the fact that is does not really have anything to do with the study at hand. - The first and second sentence of the conclusion are nearly identical.
Response5:Thank you for your suggestion.The discussion of genetic modification and exogenous addition is to lead to the conjecture that exogenous addition of 5-azacytidine can cause changes in the gene level of Monascus without changing the carbon and nitrogen source.
Comments 6: "We hypothesize that 5-azacytidine contributes to the generation of DNA methylation and that the DNA methylation of filamentous fungi can promote the increase of its metabolites." - This is both very obvious for a methylation agent and very superficial.
Response6:Thank you for your suggestion.The effect of methylation on biological growth and development is clear, but there are few studies in the field of Monascus. This experiment is mainly to supplement the lack of this aspect. In the future, we will conduct in-depth research on its mechanism at the molecular level and citrinin, the most concerned problem of Monascus.
Comments 7: Some strain names are still not in italics, e.g. in the discussion, second paragraph Aspercillus oryzae and Saccharomyces cerevisiae.
Response7: Thank you for your suggestion. We have checked the entire manuscript and made modifications to italicize the strains, with red annotations in the manuscript.